# LongFlow: Efficient KV Cache Compression for Reasoning Models

## Abstract

Reasoning models like OpenAI-o1 and DeepSeek-R1 have demonstrated strong capabilities in complex tasks such as mathematical reasoning and code generation. However, this leap in performance is achieved by generating a significantly greater number of output tokens, which dramatically increases deployment costs. The generation of extremely long sequences necessitates a longer KV cache, which in turn results in a substantial memory footprint and severe bandwidth pressure during attention computation. While there are numerous techniques to optimize KV cache, they are predominantly designed for long-input, short-output scenarios and are ineffective for the long-output nature of these reasoning models. The high computational cost of their importance estimation is severely exacerbated in long-output scenarios by the need for continuous context re-evaluation. To overcome this challenge, we introduce **LongFlow**, a novel KV cache compression method that employs an efficient importance estimation metric derived from an intermediate result in the attention computation using only the current query. This elegant design requires no auxiliary storage and adds negligible computational overhead. Furthermore, we implement a custom kernel that integrates Flash-Attention, importance estimation, and token eviction into a single, highly optimized operator to enhance system-level efficiency. Extensive experiments demonstrate that our method can achieve an 11.8x increase in throughput with an 80% compression of the KV cache while incurring negligible loss in model accuracy[1].

## 1 Introduction

The rise of Large Language Models (LLMs) (Yang et al., 2024; Grattafiori et al., 2024; Hurst et al., 2024; Liu et al., 2024a) marks a pivotal moment for artificial intelligence, demonstrating unprecedented capabilities and achieving state-of-the-art results across a wide range of tasks. Recently, a new class of **reasoning models** designed explicitly for complex reasoning has emerged (Guo et al., 2025; Jaech et al., 2024; Team et al., 2025; Yang et al., 2025; Meta, 2025). These models employ a long Chain-of-Thought (CoT) and generate extensive intermediate steps to analyze and solve complex problems. With their outstanding performance and widespread usage in demanding domains like mathematical reasoning and code generation, a significant paradigm shift has emerged for the training and deployment of LLMs, along with a new set of efficiency challenges.

Despite the significant breakthroughs of these reasoning models, their capabilities come at the cost of generating a large number of output tokens (Chen et al., 2024; Feng et al., 2025; Wang et al., 2025). This inflates the KV cache size, creating severe memory and bandwidth bottlenecks that substantially increase deployment costs. Previous research has explored KV cache compression in LLMs, but most efforts have concentrated on long-input scenarios. These existing methods often suffer from critical limitations for generative tasks: some only compress the KV cache during the prefill stage (Li et al., 2024; Cai et al., 2024; Su et al., 2025), while others introduce considerable computational overhead at each compression step and necessitate retaining additional information (Zhang et al., 2023; Guo et al., 2024). For example, the accumulation of attention scores in H2O (Zhang et al., 2023) requires approximately 15% of the time for attention computation, and an eviction step requires more time than the attention computation itself. These drawbacks create

---

[1]Code is available at https://anonymous.4open.science/r/LongFlow.

substantial efficiency issues, especially in the long-output paradigm of reasoning models, leaving an urgent requirement for more efficient KV cache compression methods.

To address this critical efficiency gap, we introduce LongFlow, an efficient KV cache compression method tailored for the long-output paradigm. The core principle of LongFlow is to compute a fast yet precise importance estimation metric for all historical tokens. It is exceptionally efficient as it is derived from an intermediate value in the standard attention calculation, thus requiring no auxiliary storage and incurring negligible computational overhead. The accuracy of this metric is theoretically grounded in its approximation of the attention output loss and empirically validated through extensive experiments. Beyond the algorithm, we introduce several system-level optimizations to maximize practical efficiency. We begin by employing a static KV cache, which pre-allocates the entire memory buffer to mitigate fragmentation and eliminate dynamic allocation overhead. Furthermore, LongFlow overwrites a single historical token at each decoding step, instead of every N steps, ensuring a consistent workload that is crucial for seamless integration into modern inference systems, such as vLLM (Kwon et al.,

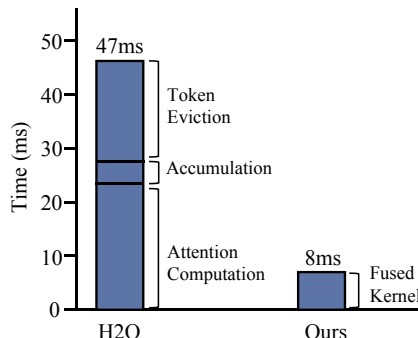

Figure 1: Time for attention module of H2O and our kernel on Qwen3-8B with a batch size of 128 and a sequence length of 3200. Both methods evict one token after attention computation.

2023) and SGLang (Zheng et al., 2024). Finally, to maximize hardware utilization, we develop a custom Triton kernel that fuses Flash-Attention, importance estimation, and token eviction into a single, high-performance operator (see speed comparison in Figure 1, the time of attention module drastically reduced from 47ms to 8ms). Extensive experiments demonstrate that our method achieves an 11.8x increase in throughput with an 80% compression ratio of the KV cache while incurring negligible loss in model accuracy.

The main contributions of our work are as follows:

- **A novel and lightweight KV cache compression algorithm.** We propose LongFlow, an algorithm designed explicitly for long-output generation. It introduces an accurate importance metric that is efficiently computed using only the current query and an intermediate result from the attention calculation, thus incurring negligible computational overhead.
- **A high-performance, fused kernel for system-level efficiency.** We design and implement a custom Triton kernel that fuses the entire attention and eviction pipeline into a single, highly-optimized operator to maximize hardware utilization.
- **State-of-the-art performance in long-output generation.** Extensive experiments demonstrate that LongFlow achieves up to a 11.8x throughput improvement, less memory fragment, and an 80% reduction in KV cache while incurring negligible loss in model accuracy.

## 2 PRELIMINARY

### 2.1 BACKGROUND: ATTENTION, KV CACHE, AND HARDWARE-AWARE OPTIMIZATION

The attention mechanism (Vaswani, 2017) is a core component of LLMs. Given an input sequence $\mathbf{X} \in \mathbb{R}^{b \times s \times d}$, where $b$ is the batch size, $s$ is the sequence length, and $d$ is the hidden dimension, it is first projected into $\mathbf{Q}, \mathbf{K}, \mathbf{V}$ by learnable weight matrices $\mathbf{W}_q, \mathbf{W}_k, \mathbf{W}_v \in \mathbb{R}^{d \times d}$. For simplicity, we omit the number of the attention heads. The attention output is computed using the Scaled Dot-Product Attention mechanism as follows:

$$\text{Attention}(\mathbf{Q}, \mathbf{K}, \mathbf{V}) = \text{softmax}\left(Mask(\frac{\mathbf{Q}\mathbf{K}^T}{\sqrt{d_k}})\right)\mathbf{V}, \tag{1}$$

where $d_k$ is the dimension of the key vectors. However, this attention mechanism relies on all past $\mathbf{K}$ and $\mathbf{V}$ when calculating the output of the current step. Therefore, LLMs typically utilize KV cache to accelerate auto-regressive decoding. The generation process of LLMs with KV cache is divided into the prefill phase and the decoding phase (Patel et al., 2024).

*i)* During the prefill phase, given input $X$, the Keys $K_{<n}$ and Values $V_{<n}$ are computed and cached. *ii)* During the decoding phase, only the Keys and Values of the new token $x_n$ need to be calculated, which are then combined with the cached Keys and Values to execute attention computation.

From Equation 1, we can find that the classical attention computation is a memory-bound operation, as it requires multiple intermediate value accesses in HBM. Fortunately, Flash-Attention Dao et al. (2022); Dao (2024) uses IO-aware operations to minimize the number of read and write operations to HBM. By processing the computation in blocks (tiling) and leveraging on-chip SRAM to store intermediate results, Flash-Attention avoids materializing the full N×N attention score matrix in HBM. This dramatically reduces memory traffic, leading to significant speedups and a lower memory footprint. Inspired by this IO-aware approach, our custom kernel also employs a block-wise computation strategy to ensure maximum efficiency, as illustrated in Figure 2.

## 2.2 Revisiting KV cache Compression in the Era of Long Reasoning Models

While prior KV cache compression techniques have demonstrated considerable success in the traditional paradigm of long-input, short-output tasks, their core design assumptions are fundamentally challenged by the rise of long reasoning models. The shift to a long-output paradigm exposes several critical limitations in these established methods:

**Prefill-Only Compression.** Many methods are confined to the prefill stage, compressing only the initial prompt (Cai et al., 2024). In long-output scenarios, where most of the tokens are generated during the decoding phase, this approach results in a drastically diminished compression ratio.

**High Computational and Memory Overhead.** Existing methods rely on token importance for eviction. The importance evaluation process is computationally expensive, adding significant latency to each compression step (Guo et al., 2024; Cai et al., 2025). Moreover, these methods often require storing auxiliary metadata, such as attention scores (Zhang et al., 2023) or past queries (Li et al., 2024), which consumes extra memory and partially cancels out the benefits of compression.

**Poor Compatibility with Modern Fused Kernels.** The performance of modern inference systems heavily relies on operator fusion like Flash-Attention (Dao et al., 2022; Dao, 2024) or Paged-Attention (Kwon et al., 2023). However, many compression algorithms are not compatible with these kernels. Their logic can interrupt the fusion process, forcing data to be moved between SRAM and HBM, or they may require recalculating intermediate results that were already available during the attention step, which can make the overall inference process even slower.

These challenges require a complete redesign of KV cache compression based on the principles of dynamic applicability, lower overhead, and system-level implementation.

## 3 Method

In this section, we introduce LongFlow, a novel KV cache compression method designed to overcome the limitations of prior work. Our approach is built on three pillars: a lightweight design philosophy, a rigorous theoretical derivation, and a high-performance system implementation.

### 3.1 A Lightweight Design Philosophy

The design of LongFlow is guided by a lightweight philosophy, standing in stark contrast to the conventional "look-back" approach of many prior methods. These methods assume that an accurate importance estimation requires aggregating historical information. As previously analyzed, this inherent dependency on historical data leads to prohibitive computational and memory overheads. To break from this costly paradigm, we built LongFlow upon two core principles:

**The Principle of Zero-History Estimation.** Our central hypothesis is that the current query, $\mathbf{q_t}$ contains sufficient information to effectively estimate the importance of all historical tokens. To validate this premise, we conduct a preliminary analysis of SnapKV (Li et al., 2024). We evaluate its performance on the LongBench benchmark (Bai et al., 2023) while reducing the size of its query observation window. The results in Figure 5a in Appendix show that the performance of SnapKV degrades only slightly as the query window gets smaller, which empirically demonstrates its remark-

able robustness. This evidence highlights the validity of the single-query method for importance estimation, forming the empirical foundation for our lightweight design.

**The Principle of Zero-Cost Estimation** The second principle guiding LongFlow's design is Zero-Cost Estimation. This principle posits that the compression mechanism should not be an expensive, standalone process, but rather an intrinsic and nearly-free byproduct of the main attention computation. This stands in contrast to prior methods that often treat the importance estimation as a distinct step performed after the attention calculation. Our method implements this principle by deriving the importance metric directly from an intermediate value of the standard attention forward pass which allows us to reuse the values that must be computed for the attention output. The reuse eliminates the need for auxiliary storage and ensures that the compression step adds no overhead.

### 3.2 Derivation of the LongFlow Importance Metric

Our derivation of the importance metric begins with a formal definition of the eviction objective. Ideally, at any given step, we should evict the token whose removal has the minimal impact on the final logits across all future generation steps. This globally optimal objective is computationally intractable. To form a tractable objective, we first simplify our goal to minimizing the impact on the immediate next step's attention output, $\mathbf{o}_{t+1}$:

$$\arg\min_i \left\| \mathbf{o}_{t+1} - \mathbf{o}_{t+1}^{(\backslash i)} \right\|^2, \tag{2}$$

where $\mathbf{o}_{t+1}^{(\backslash i)}$ is the new attention output computed without the key-value pair of token $t_i$.

Although this objective is greatly simpler, it is still an unrealizable objective as it depends on the future query $\mathbf{q}_{t+1}$. Fortunately, we find a good approximation for this objective. In our preliminary exploration, we find that the adjacent queries ($\mathbf{q}_t, \mathbf{q}_{t+1}$) have a high similarity (see Figure 5b in Appendix, similar observation can be found in Su et al. (2025)). Intuitively, since the attention output is a weighted sum of Value vectors driven by the attention scores $\mathbf{q}^T\mathbf{k}$, the strong similarity between adjacent queries implies that the overall attention distribution will be stable. This stability suggests that the impact of evicting a token at step $t$ is an excellent proxy for its impact at step $t+1$ if we omit the softmax denominator. We thus formalize our practical approximation as follows:

$$(\text{Approximation 1}) \quad \left\| \mathbf{o}_{t+1} - \mathbf{o}_{t+1}^{(\backslash i)} \right\|^2 \approx \left\| \mathbf{o}_t - \mathbf{o}_t^{(\backslash i)} \right\|^2, \quad \text{where } i < t. \tag{3}$$

A formal error analysis of this approximation is presented in Section 3.3.

Directly computing this objective for all candidate tokens remains prohibitively expensive. To derive an efficient proxy, we analyze the structure of the attention output. The exact change in the output $\mathbf{o}_t$ after evicting a historical token $t_i$ is:

$$\Delta\mathbf{o}_t = \mathbf{o}_t - \mathbf{o}_t^{(\backslash i)} = \sum_{j=0}^{t} \frac{\exp(s_t^j)}{Z} \mathbf{v}^j - \sum_{j \neq i} \frac{\exp(s_t^j)}{Z^{(\backslash i)}} \mathbf{v}^j, \tag{4}$$

where the unnormalized attention score of token $j$ is $s_t^j = \mathbf{q}_t^T \mathbf{k}^j / \sqrt{d_k}$, and $Z = \sum_{l=0}^{t} \exp(s_t^l)$ and $Z^{(\backslash i)} = \sum_{l \neq i} \exp(s_t^l)$ are the softmax denominators before and after the eviction.

The main complexity arises from the change in this denominator from $Z$ to $Z^{(\backslash i)}$. A further approximation is to omit the impact of eviction on the softmax denominator, assuming $Z \approx Z^{(\backslash i)}$. This is a simple and reasonable assumption when the number of tokens is large (we discuss the error bound of this approximation in Section 3.3). Under this approximation, the change in the output simplifies dramatically to the contribution vector of the evicted token itself:

$$(\text{Approximation 2}) \quad \Delta\mathbf{o}_t \approx \sum_{j=0}^{t} \frac{\exp(s_t^j)}{Z} \mathbf{v}^j - \sum_{j \neq i} \frac{\exp(s_t^j)}{Z} \mathbf{v}^j = \frac{\exp(s_t^i)}{Z} \mathbf{v}^i = \alpha_t^i \mathbf{v}^i, \tag{5}$$

where $\alpha_t^i$ is the attention weight of token $i$ at time step $t$.

Thus, the objective is simplified to finding the minimum $\alpha_t^i \mathbf{v}^i$. We define the final importance score using the following equation for its computational simplicity and empirical effectiveness:

$$\text{LongFlowScore}(t_i) = \left\| \alpha_t^i \mathbf{v}^i \right\|_1 = \alpha_t^i \sum_{l=1}^{d} |(\mathbf{v}^i)_l|. \tag{6}$$

The token with the minimum LongFlowScore is selected for eviction. This method is exceptionally efficient. The contribution vectors $\mathbf{c}_t^i = \alpha_t^i \mathbf{v}^i$ are a necessary intermediate result in the standard attention computation. Therefore, calculating the LongFlow score for all tokens only requires performing an additional, lightweight reduction operation on this existing intermediate tensor. This perfectly realizes our Zero-Cost Estimation principle.

## 3.3 THEORETICAL JUSTIFICATION OF THE APPROXIMATIONS

To derive the final LongFlow score from the ideal objective (from Equation 2 to Equation 6), we introduce two primary approximations: (1) using the contribution vector $\mathbf{c}_t^i$ as a proxy for the true attention output change $\Delta \mathbf{o}_t$ (the *denominator approximation*), and (2) using the current query $\mathbf{q}_t$ as a proxy for the next-step query $\mathbf{q}_{t+1}$ (the *query approximation*). We now provide a brief analysis of the error bounds for each. The detailed proofs are available in Appendix D.

The total error $\mathcal{E}_i$ for a given token $t_i$ is the difference between the ideal next-step objective and our practical score's squared magnitude:

$$\mathcal{E}_i = \left| \left\| \mathbf{o}_{t+1} - \mathbf{o}_{t+1}^{(\backslash i)} \right\|^2 - \left\| \alpha_t^i \mathbf{v}_i \right\|^2 \right|. \tag{7}$$

Using the triangle inequality, we can decompose this error into two distinct sources:

$$\mathcal{E}_i \leq \underbrace{\left| \left\| \mathbf{o}_{t+1} - \mathbf{o}_{t+1}^{(\backslash i)} \right\|^2 - \left\| \mathbf{c}_{t+1}^i \right\|^2 \right|}_{\text{Error from Denominator Approx.}} + \underbrace{\left| \left\| \mathbf{c}_{t+1}^i \right\|^2 - \left\| \mathbf{c}_t^i \right\|^2 \right|}_{\text{Error from Query Drift}}. \tag{8}$$

**Error from the Denominator Approximation.** This approximation replaces the true output change with the contribution vector of the evicted token. A rigorous algebraic manipulation shows that the remainder term of this approximation, $\mathbf{R}_{t+1}^i = \Delta \mathbf{o}_{t+1} - \mathbf{c}_{t+1}^i$, is bounded by $\| \mathbf{R}_{t+1}^i \| \leq \frac{2V \alpha_{t+1}^i}{1 - \alpha_{t+1}^i}$, where $V = \max_j \| \mathbf{v}^j \|$. This bound confirms that the approximation is highly accurate when the attention weight of the evicted token $\alpha_t^i$ is small. Since our method is designed to evict tokens with low attention, this approximation is well-justified.

**Error from the Query Approximation.** Query approximation uses the score computed with the current query as a proxy for the ideal score that would be computed with the future query, $\mathbf{q}_{t+1}$. The error introduced by this temporal approximation originates from the difference in attention weights, $|\alpha_{t+1}^i - \alpha_t^i|$. Due to the Lipschitz continuity of the softmax function, the change in output weights is bounded by the maximum change in the input pre-softmax scores, $s_t^j$:

$$|\alpha_{t+1}^i - \alpha_t^i| \leq \max_j |s_{t+1}^j - s_t^j|. \tag{9}$$

The error in the scores, $\Delta s^j = s_{t+1}^j - s_t^j$, arises from the query shift. By the Cauchy-Schwarz inequality, this error is bounded by $|\Delta s^j| \leq \frac{\|\mathbf{q}_{t+1} - \mathbf{q}_t\| \cdot \|\mathbf{k}^j\|}{\sqrt{d_k}}$. Assuming queries are normalized to unit vectors, we can relate the query difference to their cosine similarity: $\|\mathbf{q}_{t+1} - \mathbf{q}_t\|^2 = 2(1 - \cos(\mathbf{q}_t, \mathbf{q}_{t+1}))$. This gives a final bound on the maximum score error:

$$\max_j |\Delta s^j| \leq \frac{\sqrt{2(1 - \cos(\mathbf{q}_t, \mathbf{q}_{t+1}))} \cdot \max_j \|\mathbf{k}^j\|}{\sqrt{d_k}}. \tag{10}$$

This bound provides a strong theoretical guarantee. It shows that as the similarity between consecutive queries approaches 1, the error from our temporal approximation approaches 0.

## 3.4 HIGH-PERFORMANCE SYSTEM IMPLEMENTATION

To realize the theoretical efficiency of LongFlow as practical speedups, we implement a high-performance system centered around a custom fused kernel. This section details our key system-level optimizations, including the memory management strategy and the fused attention & eviction kernel. Figure 2 shows the data and computation flow of our method.

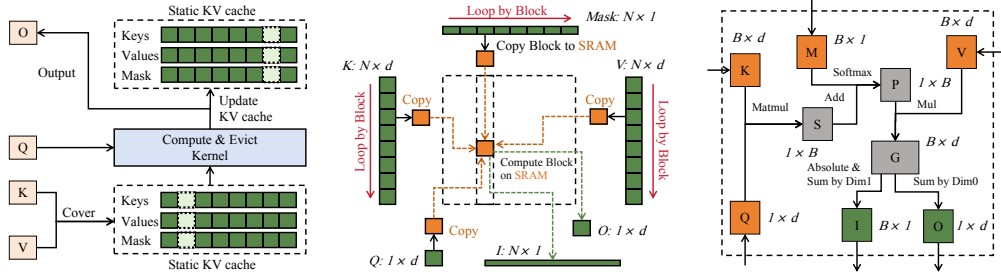

Figure 2: The data and computation flow of our method. O: attention output; I: LongFlowScore; S, P and G are intermediate states in kernel forward pass. (**Left**): The process of a decoding step. The current KV will cover a slot selected in the previous step, and then the static KV and Mask will be sent to the kernel together with Q for calculation to obtain the current attention output and the slot to be covered in the next step. (**Middle**): The data flow between HBM and SRAM in the kernel. KV and Mask will enter SRAM by block and perform fused attention calculation. (**Right**): The computational flow on chip. Unlike standard flash attention calculations, we split the matrix multiplication of P and V into two steps and derive LongFlowScore from the intermediate result G.

**Static Memory and Consistent Workload.** As established in our design philosophy, we employ a static KV cache and a consistent per-step eviction policy. The entire memory for the KV cache is pre-allocated to eliminate dynamic allocation overheads and memory fragmentation. By overwriting a single token at every decoding step, we ensure a consistent computational workload. This predictability is crucial for simplifying integration into modern inference engines(Kwon et al., 2023; Zheng et al., 2024). Furthermore, this operational consistency is particularly beneficial in distributed systems, as it guarantees a balanced workload and predictable overhead across all parallel workers.

**Fused Attention and Eviction Kernel.** The core of our implementation is a custom Triton kernel that fuses the entire attention and eviction process. While our kernel is designed based on the I/O-aware principles of FlashAttention (Dao et al., 2022), it incorporates three critical modifications to specifically and efficiently handle the auto-regressive decoding process with LongFlow: (1) Given that the query sequence length is always 1 during decoding, we eliminate the outer loop over the query dimension in standard FlashAttention. This simplifies the design to a single, highly efficient pass over the historical KV cache by block. (2) To seamlessly integrate the online calculation of the LongFlowScore, we omit the running maximum used for numerical stability in safe softmax. To compensate and prevent potential overflow, we perform the calculation of softmax in FP32. (3) We restructure the computation to maximize data reuse for both tasks. Within the single loop over KV cache blocks, we compute the un-normalized contribution vectors $exp(scores)V$ as a key intermediate result. This tensor is then summed along the sequence dimension to update the final attention output accumulator, and its L1-norm is calculated to serve as the un-normalized LongFlowScore for that block. These modifications result in a single, highly-optimized operator that computes both the attention output and the token to evict in one pass, as detailed in Algorithm 1 in Appendix.

## 4 EXPERIMENTS

We conduct a series of experiments to comprehensively evaluate the effectiveness of LongFlow. Our evaluation focuses on three key aspects: model quality, inference speed, and memory efficiency.

### 4.1 EXPERIMENTAL SETUP

**Models.** We evaluate LongFlow on different reasoning models to demonstrate its applicability. Our experiments are primarily conducted on **DeepSeek-R1-Distill-Llama-8B** (Guo et al., 2025) and the **Qwen3** (Yang et al., 2025) series, including its 0.6B, 1.7B, 4B, and 8B variants.

**Datasets.** Our evaluation spans a wide spectrum of challenging reasoning benchmarks to ensure a comprehensive assessment of model accuracy. For competition-level mathematics, we use problems from MATH-500 (Hendrycks et al., 2021), AMC-23 (MAA, 2023), AIME-24 (MAA, 2024) and AIME-25 (MAA, 2025). To test advanced scientific reasoning, we employ the graduate-level expert

Table 1: Model Performance on different models across different datasets. Numbers in parentheses indicate dataset sizes. **Bold** indicates the best performance.

| Model | Method | AIME24 (30) | AIME25 (30) | AMC (40) | GPQA (198) | GSM8K (1318) | MATH (500) | Minerva (272) | Olympiad (675) | Avg |
|---|---|---|---|---|---|---|---|---|---|---|
| | | | | | **Budget = 2400** | | | | | |
| | Vanilla | 30.00 | 20.00 | 77.50 | 41.41 | 77.48 | 83.40 | 20.59 | 44.30 | 62.73 |
| | H2O | **43.33** | **20.00** | 65.00 | **41.41** | **77.71** | 79.80 | 19.12 | 44.00 | 62.01 |
| | R-KV | 36.67 | 20.00 | **75.00** | 38.38 | 77.41 | **80.80** | 20.22 | **45.93** | **62.43** |
| | VATP | 33.33 | 13.33 | 72.50 | 33.84 | 77.48 | 79.80 | **20.59** | 43.56 | 61.39 |
| DeepSeek-R1 | Ours | 33.33 | 16.67 | 70.00 | 40.40 | 77.71 | 79.80 | 20.22 | 43.85 | 61.95 |
| -Distill1 | | | | | **Budget = 3200** | | | | | |
| -Llama-8B | Vanilla | 30.00 | 20.00 | 77.50 | 41.41 | 77.47 | 83.40 | 20.59 | 44.30 | 62.72 |
| | H2O | 36.67 | 23.33 | 72.50 | 38.38 | **77.69** | 81.60 | **21.32** | 46.22 | 62.88 |
| | R-KV | 30.00 | 23.33 | 77.50 | **42.93** | 77.62 | **82.40** | 19.49 | **46.81** | **63.20** |
| | VATP | **40.00** | 16.67 | 72.50 | 42.42 | 77.39 | 81.60 | 19.12 | 46.67 | 62.88 |
| | Ours | 33.33 | **26.67** | **82.50** | 40.91 | 77.62 | 79.80 | 20.22 | 45.93 | 62.64 |
| | | | | | **Budget = 2400** | | | | | |
| | Vanilla | 60.00 | 46.67 | 90.00 | 34.85 | 95.68 | 92.60 | 33.46 | 58.22 | 76.57 |
| | H2O | 40.00 | 20.00 | 65.00 | 26.77 | 95.98 | 85.20 | 32.35 | 49.04 | 72.06 |
| | R-KV | 33.33 | 23.33 | 85.00 | 28.79 | 96.13 | **89.00** | 32.72 | 53.33 | **74.09** |
| | VATP | **40.00** | 26.67 | 77.50 | 23.23 | **96.29** | 87.60 | 31.99 | 49.33 | 72.62 |
| | Ours | 36.67 | **26.67** | 80.00 | **30.81** | 95.83 | 88.00 | 31.25 | 51.11 | 73.30 |
| Qwen3-8B | | | | | **Budget = 3200** | | | | | |
| | Vanilla | 60.00 | 46.67 | 90.00 | 34.85 | 95.68 | 92.60 | 33.46 | 58.22 | 76.56 |
| | H2O | 46.67 | 26.67 | 80.00 | 27.78 | 96.36 | 90.40 | **35.29** | 53.78 | 74.76 |
| | R-KV | 50.00 | **33.33** | 82.50 | **32.32** | 96.36 | **91.60** | 33.46 | **55.56** | **75.61** |
| | VATP | 43.33 | 33.33 | 82.50 | 30.30 | **96.43** | 91.00 | 34.19 | 55.41 | 75.38 |
| | Ours | **50.00** | 26.67 | **85.00** | 30.30 | 96.36 | 89.80 | 34.19 | 55.56 | 75.22 |

QA from GPQA (Rein et al., 2024), university-level problems from Minerva (Lewkowycz et al., 2022), and Olympiad-level questions from OlympiadBench (He et al., 2024). Finally, we include the widely-used GSM8K benchmark (Cobbe et al., 2021) for grade-school math word problems.

**Baselines.** We compare LongFlow against several representative KV cache compression methods:

- **Vanilla:** The standard attention without any KV cache management. It serves as the accuracy upper bound but represents the worst scenario for memory consumption and latency.
- **H2O**(Zhang et al., 2023): A classic importance-based method that prioritizes tokens based on accumulated attention scores and recency.
- **VATP**(Guo et al., 2024): An advanced method that integrates both attention scores and Value vector information into its importance metric.
- **R-KV**(Cai et al., 2025): A recent method also designed for reasoning models. It incorporates information about repetitive patterns in the generation to guide its eviction policy, with a primary focus on maximizing model accuracy over inference speed.

**Evaluation Metrics.** We evaluate all methods across two primary dimensions. For system performance, we measure the inference throughput, peak GPU memory usage, and memory fragments. For model accuracy, we report the task-specific accuracy on each of the benchmarks.

**Implementation Details.** For our main experiments, the generation output length is set to 16,000 tokens. For all compression methods, we set the KV cache budget to be either 3,200 or 2,400 tokens. For H2O and VATP, this budget is evenly split between heavy-hitter tokens and recent tokens. For LongFlow, if the number of tokens in the prefill stage exceeds the budget, we first use SnapKV (Li et al., 2024) to compress the prefill KV cache down to the target budget size. For all the baselines, we set the compression interval to achieve a balance between compression ratio and inference speed. All the experiments are conducted on NVIDIA A100 40GB GPUs.

## 4.2 MAIN RESULTS ON MODEL ACCURACY

We present our main results on model accuracy in Table 1. The evaluation across two distinct model families and two KV cache budget settings demonstrates that LongFlow effectively preserves the reasoning capabilities of the base models, achieving performance that is highly competitive with state-of-the-art baselines and close to the uncompressed Vanilla model.

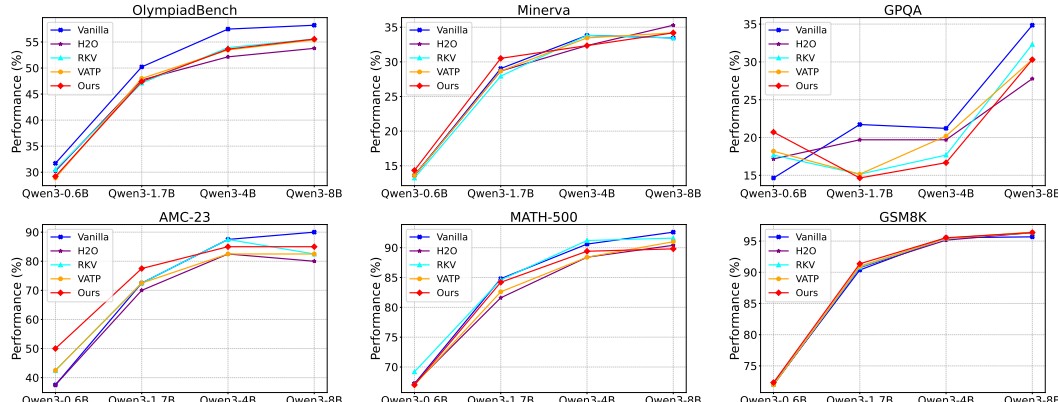

Figure 3: Accuracy of LongFlow and the baselines across different model sizes on different datasets.

**Comparison With the Baselines.** Across both models and budget settings, our proposed LongFlow demonstrates highly competitive performance against state-of-the-art baselines. R-KV generally achieves the highest average accuracy (with more memory for redundant token identification), establishing itself as a strong baseline focused on quality preservation. LongFlow consistently performs on par with or better than other methods like VATP and H2O.

**Comparison with Full KV and Different Cache Budgets.** A key finding is that all compression methods, including LongFlow, successfully preserve the vast majority of the original model's capabilities while operating on a significantly reduced KV cache (e.g., a 3.2k budget representing an 80% compression ratio for a 16k generation). As shown in Table 1, the average performance degradation compared to the uncompressed Vanilla baseline is negligible to minor—as low as 0.08% for DeepSeek-R1-Distill-Llama-8B and approximately 1.3% for Qwen3-8B. When the KV cache budget is tightened from 3.2k to 2.4k, a graceful degradation is observed across all methods. LongFlow performs better under this pressure, showing less degradation than both H2O and VATP.

### 4.3 PERFORMANCE ON DIFFERENT MODEL SIZES

To assess the performance of LongFlow across different model sizes, we conduct a comparative analysis using the Qwen3 model family, including the 0.6B, 1.7B, 4B, and 8B variants. For this specific analysis, the AIME-24 and AIME-25 datasets were excluded due to their small sample sizes and for brevity. The results in Figure 3 show that LongFlow maintains a highly competitive performance, achieving results that are comparable or superior to the baselines across all tested model sizes and benchmarks, which demonstrates the effectiveness and robustness of our method.

### 4.4 THROUGHPUT AND MEMORY ANALYSIS

Finally, we evaluate the system-level performance of LongFlow in terms of throughput and memory efficiency. For this analysis, we use the Qwen3-1.7B on a single NVIDIA A100 40GB GPU. We configure a scenario with an input length of 512 and an output length of 16,000. We set the KV cache budget to 3,200 for all compression methods. To measure the maximum performance, we progressively increase the batch size for each method until an Out-of-Memory (OOM) error occurs. We report the throughput, peak memory footprint, and memory fragments in Figure 4.

The results clearly demonstrate LongFlow's substantial advantages in both throughput and memory management. In terms of throughput, LongFlow outperforms FullKV by a remarkable 11.8 times and is approximately 4.0 times faster than other methods. Regarding peak memory usage, LongFlow's footprint is comparable to other methods under the same budget, while R-KV consumes slightly more due to its more complex importance estimation logic. Notably, LongFlow's advantage in memory management lies in its superior handling of fragmentation. Its static memory scheme and consistent per-step eviction policy result in a memory layout that is both less fragmented and highly predictable. This less fragmented memory layout allows LongFlow to support a larger maximum batch size than competing methods. The predictability makes LongFlow more stable and reliable in real-world deployment by eliminating runtime memory-related uncertainties.

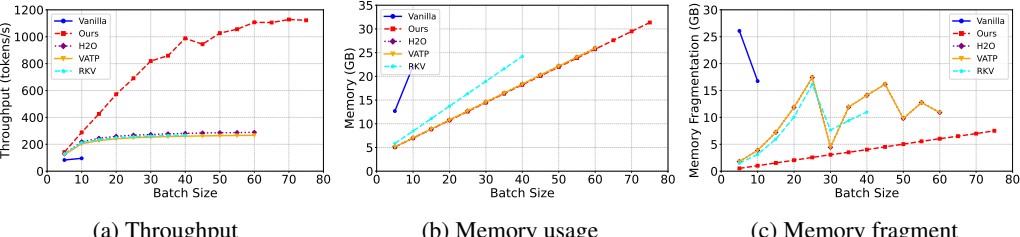

(a) Throughput        (b) Memory usage        (c) Memory fragment

Figure 4: Performance comparison of LongFlow against baselines. The compression is conducted every step for LongFlow and every 128 steps for other methods. LongFlow achieves higher throughput and supports a larger maximum batch size due to superior memory management.

## 5 RELATED WORKS

**Reasoning Models** While traditional LLMs perform well on general tasks, achieving breakthroughs in complex logical reasoning has remained a significant challenge. This bottleneck has recently been addressed by a new class of models named reasoning models designed for long CoT generation. Pioneered by OpenAI-o1 (Jaech et al., 2024), DeepSeek-R1 (Guo et al., 2025) leverages Reinforcement Learning with Verifiable Rewards (RLVR) to unlock the model's latent abilities for long CoT generation and high-level reasoning. These breakthroughs catalyze a subsequent wave of increasingly powerful reasoning models from numerous research labs, including Qwen3 (Yang et al., 2025), Gemini-2.5 (Comanici et al., 2025), GPT-5 (OpenAI, 2025), and Claude-4 (Anthropic, 2025). Researchers have employed advanced training strategies and efficient training systems to develop increasingly powerful models that mark new milestones in history. However, the long-CoT reasoning paradigm introduces significant deployment challenges. Generating lengthy sequences inflates the KV cache, leading to severe memory and computational bottlenecks. This problem is further exacerbated by the models' tendency to overthink (Chen et al., 2024). While numerous approaches have been proposed to develop more efficient reasoning models (Feng et al., 2025; Wang et al., 2025; Jiang et al., 2025), they often focus on shortening the CoT, which risks constraining the model's reasoning capability. Developing more effective methods remains an open challenge.

**KV Cache Compression** Managing the growing KV cache is a central challenge in efficient LLM inference, with a diverse range of strategies proposed to retain critical context within a fixed memory budget, named KV cache compression. Positional strategies, like StreamingLLM (Xiao et al., 2023), preserve a fixed window of recent tokens alongside crucial initial "attention sinks." In parallel, importance-based methods have evolved from using simple heuristics like accumulated attention scores in H2O (Zhang et al., 2023) and SnapKV (Li et al., 2024), to more direct objectives like optimizing attention output similarity in AdaKV (Feng et al., 2024). While these eviction-centric methods are widely studied, their core drawback is the irreversible information loss from permanently discarding tokens. To mitigate this, more sophisticated techniques have emerged that create compact representations rather than simply pruning. These include semantic consolidation via chunking in ChunkKV (Liu et al., 2025), and cross-layer merging in MiniCache (Liu et al., 2024b). A distinct, dynamic approach is seen in RefreshKV (Xu et al., 2024), which periodically refreshes the cache to prevent important context from being lost permanently. Although these innovations are powerful, they often introduce substantial computational overhead or architectural complexity, highlighting the persistent need for a solution that is both effective and fundamentally lightweight.

## 6 CONCLUSION

This paper introduces LongFlow, a novel and lightweight KV cache compression method designed to mitigate the significant inference overhead in long-output reasoning models. Grounded in a zero-history and zero-cost design philosophy, LongFlow employs a highly efficient, theoretically-justified importance metric derived directly from intermediate attention values, incurring negligible computational cost. With our co-designed system, Longflow increase throughput by up to 11.8x and reduce the KV cache footprint by 80%, while maintaining high model accuracy. By striking an effective balance between performance and accuracy, LongFlow presents a practical pathway toward the efficient deployment and broader accessibility of next-generation reasoning models.

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

## A   ETHICS STATEMENT

We confirm that this work adheres to ethical research practices. All data and LLMs used are publicly available (including API format) and properly cited.

## B   REPRODUCIBILITY STATEMENT

Comprehensive details of the experimental settings, hyper parameters, and evaluation details are presented in Section 4.1. The implementation, including source code and execution scripts, is anonymously released.

## C   USE OF LLMS

During the writing of this paper, we leverage large language models (LLMs) to refine the clarity and fluency of our writing, particularly in the Abstract and Introduction sections. Specifically, we used the Qwen web interface [2] to access the Qwen series of models (e.g., Qwen-Max), inputting early drafts of these sections and requesting stylistic improvements while preserving technical accuracy and original intent. The model's suggestions helped enhance sentence structure, academic tone, and overall readability. All final content was carefully reviewed, validated, and edited by the authors to ensure fidelity to our research and adherence to scholarly standards.

## D   DETAILED ERROR BOUND DERIVATION

In this section, we provide the detailed proofs for the two key approximations that underpin the LongFlow method, justifying the claims made in Section 3.3. Our goal is to bound the total error $\mathcal{E}^i = \left| \left\| \mathbf{o}_{t+1} - \mathbf{o}_{t+1}^{(\backslash i)} \right\|^2 - \left\| \mathbf{c}_t^i \right\|^2 \right|$, which we decompose using the triangle inequality:

$$\mathcal{E}^i \leq \underbrace{\left| \left\| \mathbf{o}_{t+1} - \mathbf{o}_{t+1}^{(\backslash i)} \right\|^2 - \left\| \mathbf{c}_{t+1}^i \right\|^2 \right|}_{\text{Error from Denominator Approx.}} + \underbrace{\left| \left\| \mathbf{c}_{t+1}^i \right\|^2 - \left\| \mathbf{c}_t^i \right\|^2 \right|}_{\text{Error from Query Drift}}. \tag{11}$$

We now derive explicit bounds for each of these two error terms.

### D.1   BOUNDING THE ERROR FROM THE DENOMINATOR APPROXIMATION

This error arises from approximating the true change in attention output, $\Delta \mathbf{o}_{t+1} = \mathbf{o}_{t+1} - \mathbf{o}_{t+1}^{(\backslash i)}$, with the contribution vector $\mathbf{c}_{t+1}^i = \alpha_{t+1}^i \mathbf{v}^i$. We now derive an exact expression for the remainder term $\mathbf{R}_{t+1}^i = \Delta \mathbf{o}_{t+1} - \mathbf{c}_{t+1}^i$.

First, let us expand the terms. The original attention output is a sum over all $t$ historical tokens, while the output after eviction is a sum over the remaining $t - 1$ tokens with re-normalized attention weights:

$$\mathbf{o}_{t+1} = \sum_{j=1}^{t} \alpha_{t+1}^j \mathbf{v}^j = \alpha_{t+1}^i \mathbf{v}^i + \sum_{j \neq i} \alpha_{t+1}^j \mathbf{v}^j$$

$$\mathbf{o}_{t+1}^{(\backslash i)} = \sum_{j \neq i} \alpha_{t+1}^{j,(\backslash i)} \mathbf{v}^j$$

---

[2] https://chat.qwen.ai

Substituting these into the definition of the remainder term:

$$
\begin{aligned}
\mathbf{R}_{t+1}^i &= \Delta \mathbf{o}_{t+1} - \mathbf{c}_{t+1}^i \\
&= \left( \mathbf{o}_{t+1} - \mathbf{o}_{t+1}^{(\backslash i)} \right) - \mathbf{c}_{t+1}^i \\
&= \left( \left( \alpha_{t+1}^i \mathbf{v}^i + \sum_{j \neq i} \alpha_{t+1}^j \mathbf{v}^j \right) - \left( \sum_{j \neq i} \alpha_{t+1}^{j,(\backslash i)} \mathbf{v}^j \right) \right) - \alpha_{t+1}^i \mathbf{v}^i \\
&= \sum_{j \neq i} \left( \alpha_{t+1}^j - \alpha_{t+1}^{j,(\backslash i)} \right) \mathbf{v}^j.
\end{aligned}
$$

The key is to relate the new weights $\alpha_{t+1}^{j,(\backslash i)}$ to the old weights $\alpha_{t+1}^j$. The new weights are re-normalized over a smaller set of tokens:

$$
\alpha_{t+1}^{j,(\backslash i)} = \frac{\exp(s_{t+1}^j)}{\sum_{l \neq i} \exp(s_{t+1}^l)} = \frac{\exp(s_{t+1}^j)}{Z_{t+1} - \exp(s_{t+1}^i)} = \frac{\exp(s_{t+1}^j)/Z_{t+1}}{(Z_{t+1} - \exp(s_{t+1}^i))/Z_{t+1}} = \frac{\alpha_{t+1}^j}{1 - \alpha_{t+1}^i}.
$$

Now, we substitute this identity back into the expression for the remainder term:

$$
\mathbf{R}_{t+1}^i = \sum_{j \neq i} \left( \alpha_{t+1}^j - \frac{\alpha_{t+1}^j}{1 - \alpha_{t+1}^i} \right) \mathbf{v}^j = \sum_{j \neq i} \left( \frac{-\alpha_{t+1}^j \alpha_{t+1}^i}{1 - \alpha_{t+1}^i} \right) \mathbf{v}^j = -\frac{\alpha_{t+1}^i}{1 - \alpha_{t+1}^i} \sum_{j \neq i} \alpha_{t+1}^j \mathbf{v}^j.
$$

Finally, we recognize that the remaining summation is simply the original attention output minus the contribution of the $i$-th token: $\sum_{j \neq i} \alpha_{t+1}^j \mathbf{v}^j = \mathbf{o}_{t+1} - \mathbf{c}_{t+1}^i$. This gives the final exact expression for the remainder term:

$$
\mathbf{R}_{t+1}^i = -\frac{\alpha_{t+1}^i}{1 - \alpha_{t+1}^i} \left( \mathbf{o}_{t+1} - \mathbf{c}_{t+1}^i \right). \tag{12}
$$

Taking the norm and assuming $\|\mathbf{v}^j\| \leq V$, we have $\|\mathbf{o}_{t+1}\| \leq V$. The norm of the remainder is then bounded:

$$
\|\mathbf{R}_{t+1}^i\| \leq \frac{\alpha_{t+1}^i}{1 - \alpha_{t+1}^i} \left( \|\mathbf{o}_{t+1}\| + \|\mathbf{c}_{t+1}^i\| \right) \leq \frac{\alpha_{t+1}^i (1 + \alpha_{t+1}^i)}{1 - \alpha_{t+1}^i} V \leq \frac{2V \alpha_{t+1}^i}{1 - \alpha_{t+1}^i}. \tag{13}
$$

### D.2 BOUNDING THE ERROR FROM THE QUERY APPROXIMATION

This error arises from query drift, captured by the term $\left| \|\mathbf{c}_{t+1}^i\|^2 - \|\mathbf{c}_t^i\|^2 \right| = \left| (\alpha_{t+1}^i)^2 - (\alpha_t^i)^2 \right| \cdot \|\mathbf{v}^i\|^2$. The core task is to bound $|\alpha_{t+1}^i - \alpha_t^i|$.

**Lemma 1** (Lipschitz Property of Softmax). *The softmax function $\sigma(\mathbf{s})_i = \exp(s^i)/\sum_j \exp(s^j)$ is 1-Lipschitz continuous with respect to the L1-norm on its output and the L-infinity norm on its input.*

*Proof.* By the Mean Value Theorem for vector-valued functions, for any two score vectors $\mathbf{s}$ and $\mathbf{s}'$, there exists a point $\mathbf{c}$ on the line segment between them such that:

$$
\sigma(\mathbf{s}') - \sigma(\mathbf{s}) = J_\sigma(\mathbf{c})(\mathbf{s}' - \mathbf{s})
$$

where $J_\sigma$ is the Jacobian matrix of the softmax function. Our goal is to bound the norm of this Jacobian. The entries of the Jacobian, $J_{ij} = \frac{\partial \sigma_i}{\partial s^j}$, are computed as follows:

For the diagonal entries ($i = j$):

$$
\begin{aligned}
\frac{\partial \sigma_i}{\partial s^i} &= \frac{\partial}{\partial s^i} \left( \frac{\exp(s^i)}{\sum_k \exp(s^k)} \right) \\
&= \frac{\exp(s^i) \left( \sum_k \exp(s^k) \right) - \exp(s^i) \exp(s^i)}{\left( \sum_k \exp(s^k) \right)^2} \\
&= \frac{\exp(s^i)}{\sum_k \exp(s^k)} \left( 1 - \frac{\exp(s^i)}{\sum_k \exp(s^k)} \right) = \sigma_i(1 - \sigma_i).
\end{aligned}
$$

For the off-diagonal entries ($i \neq j$):

$$\frac{\partial \sigma_i}{\partial s^j} = \frac{\partial}{\partial s^j} \left( \frac{\exp(s^i)}{\sum_k \exp(s^k)} \right)$$

$$= \frac{0 - \exp(s^i)\exp(s^j)}{(\sum_k \exp(s^k))^2}$$

$$= -\frac{\exp(s^i)}{\sum_k \exp(s^k)} \frac{\exp(s^j)}{\sum_k \exp(s^k)} = -\sigma_i \sigma_j.$$

Thus, the Jacobian can be succinctly written as $J_{ij} = \sigma_i(\delta_{ij} - \sigma_j)$, where $\delta_{ij}$ is the Kronecker delta.

To obtain the bound used in the main text, we need to bound the induced L1/L$\infty$ norm. It is a known property of the softmax function that $\|\sigma(\mathbf{s}') - \sigma(\mathbf{s})\|_1 \leq \|\mathbf{s}' - \mathbf{s}\|_\infty$, which means the Lipschitz constant for this norm pairing is 1. While the full proof of this specific norm bound is lengthy and typically presented in optimization literature, we can gain insight by analyzing the simpler induced L1-norm, $\|J\|_1 = \max_j \sum_i |J_{ij}|$. The sum of the absolute values of the $j$-th column is:

$$\sum_i |J_{ij}| = |\sigma_j(1 - \sigma_j)| + \sum_{i \neq j} |-\sigma_i \sigma_j|$$

$$= \sigma_j(1 - \sigma_j) + \sum_{i \neq j} \sigma_i \sigma_j \quad (\text{since } \sigma_k \in [0, 1])$$

$$= \sigma_j - \sigma_j^2 + \sigma_j \sum_{i \neq j} \sigma_i$$

$$= \sigma_j - \sigma_j^2 + \sigma_j(1 - \sigma_j) = 2\sigma_j(1 - \sigma_j).$$

Since the maximum value of the function $f(x) = 2x(1 - x)$ for $x \in [0, 1]$ is $1/2$ (at $x = 1/2$), the induced L1-norm is bounded by $\|J\|_1 \leq 1/2$. The bound of 1 for the L1/$L^\infty$ norm pairing used in our main analysis is a tighter, more specific result. $\square$

Applying the Lemma, the change in a single attention weight is bounded by the maximum change in any pre-softmax score:

$$|\alpha_{t+1}^i - \alpha_t^i| \leq \max_j |s_{t+1}^j - s_t^j|. \tag{14}$$

The error in the scores, $s_{t+1}^j - s_t^j = (\mathbf{q}_{t+1} - \mathbf{q}_t)^T \mathbf{k}^j / \sqrt{d_k}$, is bounded by the Cauchy-Schwarz inequality. Assuming unit-normalized queries, we have $\|\mathbf{q}_{t+1} - \mathbf{q}_t\|^2 = 2(1 - \cos(\mathbf{q}_t, \mathbf{q}_{t+1}))$. This gives the final bound on the maximum score error:

$$\max_j |\Delta s^j| \leq \frac{\sqrt{2(1 - \cos(\mathbf{q}_t, \mathbf{q}_{t+1}))} \cdot \max_j \|\mathbf{k}^j\|}{\sqrt{d_k}}. \tag{15}$$

### D.3 Combined Error Bound

By combining the bounds for both error sources, we can conclude that the total error $\mathcal{E}^i$ is small when our method evicts a token with a small attention weight $\alpha_{t+1}^i$ and when the query similarity is high. This provides a rigorous theoretical foundation for the effectiveness of the LongFlow method.

## E    Algorithm of LongFlow Kernel

Algorithm 1 shows the workflow of our kernel.

## F    Additional Figures

Figure 5 shows the figures we used in our preliminary experiments.

**Algorithm 1** LongFlow Fused Attention and Eviction Kernel

1: **Input:** Query $\mathbf{q}_t \in \mathbb{R}^{1 \times d}$, KV cache $\mathbf{K} \in \mathbb{R}^{(t-1) \times d}$, $\mathbf{V} \in \mathbb{R}^{(t-1) \times d}$, Mask $\mathbf{M} \in \{0,1\}^{t-1}$
2: **Output:** Attention output $\mathbf{o}_t \in \mathbb{R}^{1 \times d}$, Eviction index $i_{\text{evict}}$
3:
4: *// Initialize accumulators*
5: $\mathbf{o}_{\text{acc}} \leftarrow \mathbf{0} \in \mathbb{R}^d$;                                  ▷ Output accumulator
6: $l_{\text{acc}} \leftarrow 0 \in \mathbb{R}$                                  ▷ Softmax denominator accumulator
7: $\mathbf{S}_{\text{loss}} \leftarrow \mathbf{0} \in \mathbb{R}^{t-1}$                                  ▷ Temporary storage for un-normalized scores
8: $B_k \leftarrow \text{block\_size}$
9:
10: *// Single pass over the KV cache in blocks*
11: **for** $j \leftarrow 1, B_k, \ldots, t-1$ **do**
12:     *// Load a block of K, V from HBM to on-chip SRAM*
13:     $\mathbf{K}_j \leftarrow \mathbf{K}[j : j + B_k, :]$;     $\mathbf{V}_j \leftarrow \mathbf{V}[j : j + B_k, :]$
14:
15:     *// Compute scores and apply mask for the current block*
16:     $\mathbf{S}_j \leftarrow \mathbf{q}_t \mathbf{K}_j^T / \sqrt{d}$
17:     $\mathbf{S}_j[\neg \mathbf{M}_j] \leftarrow -\infty$
18:
19:     *// Compute softmax numerators and update denominator accumulator*
20:     $\mathbf{P}_j \leftarrow \exp(\mathbf{S}_j)$
21:     $l_{\text{acc}} \leftarrow l_{\text{acc}} + \text{sum}(\mathbf{P}_j)$
22:
23:     *// Compute un-normalized contribution vectors and update output accumulator*
24:     $\mathbf{C}_j \leftarrow \mathbf{P}_j \cdot \mathbf{V}_j$                                  ▷ Element-wise scaling, shape: $B_k \times d$
25:     $\mathbf{o}_{\text{acc}} \leftarrow \mathbf{o}_{\text{acc}} + \text{sum}(\mathbf{C}_j, \dim=0)$
26:
27:     *// Compute and store un-normalized importance for the block*
28:     $\mathbf{s}_{\text{loss},j} \leftarrow \text{sum}(|\mathbf{C}_j|, \dim=1)$                                  ▷ L1-norm of contribution vectors
29:     Store $\mathbf{s}_{\text{loss},j}$ at corresponding indices in $\mathbf{S}_{\text{loss}}$
30: **end for**
31:
32: *// Post-processing after the loop*
33: $\mathbf{o}_t \leftarrow \mathbf{o}_{\text{acc}} / l_{\text{acc}}$                                  ▷ Normalize the final attention output
34: $\mathbf{S}_{\text{final\_loss}} \leftarrow \mathbf{S}_{\text{loss}} / l_{\text{acc}}$                                  ▷ Normalize the importance scores
35: $i_{\text{evict}} \leftarrow \text{argmin}(\mathbf{S}_{\text{final\_loss}})$                                  ▷ Find the token with the minimum score
36:
37: **return** $\mathbf{o}_t, i_{\text{evict}}$

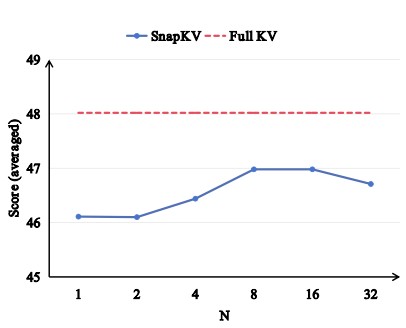
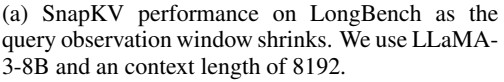

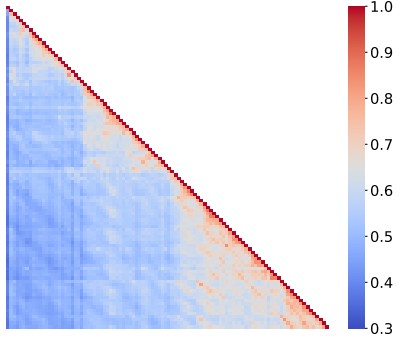

(a) SnapKV performance on LongBench as the query observation window shrinks. We use LLaMA-3-8B and an context length of 8192.

(b) Cosine similarity between queries. Results are shown for the first 100 tokens, averaged over 100 samples from MATH500 using Qwen3-8B (Layer 10, Head 10). The high and stable similarity between adjacent queries supports our approximation.

Figure 5: Empirical motivation for our single-query hypothesis.

