# OpenReview forum: "LongFlow: Efficient KV Cache Compression for Reasoning Models"
_ICLR.cc/2026/Conference — ICLR 2026 Conference Withdrawn Submission_

### Official Review · Reviewer_NEao · 2025-10-26

**Soundness:** 1
**Presentation:** 1
**Contribution:** 2
**Rating:** 2
**Confidence:** 5

**Summary:**

The paper proposes a KV eviction method for reasoning models, termed LongFlow. Specifically, it addresses long-decoding scenarios, offering improvements over previous prefill-based compression methods. The primary goal of the proposed approach is to preserve future attention output after KV eviction, which the authors approximate using current query features. The paper also presents theoretical error bounds for these approximations. In addition, the authors develop a fused Triton kernel implementation of the KV eviction algorithm, significantly enhancing throughput compared to earlier naive implementations. The proposed method demonstrates comparable performance on reasoning benchmarks when evaluated with distilled LLaMA3-8B and Qwen3 model series.

**Strengths:**

- The paper addresses a timely and important problem that is of significant interest to the ICLR community, particularly in the context of efficient reasoning.
- The authors develop a fused Triton kernel implementation that achieves higher throughput compared to previous KV eviction implementations, demonstrating strong practical relevance.

**Weaknesses:**

Insufficient justification and insight for the proposed approximation
- The simplification at L178 (“To form a tractable objective, we first simplify our goal to minimizing the impact on the immediate next step’s attention output”) would benefit from stronger justification. The gap between optimizing for all future steps and focusing on a single-step approximation could be better supported through statistical evidence or ablation studies.
- The observation of high similarity between adjacent queries is currently supported by limited evidence. The appendix figure illustrates this for a single example; expanding this analysis with statistical validation using more data and models would strengthen the argument.
- The theoretical justification also remains somewhat intuitive (e.g., “if queries are similar, then attention outputs are similar”). Including empirical analysis of approximation errors or additional ablation studies of approximations could provide stronger support for the paper’s contribution.

Weak empirical results and presentation
- Table 1 is informative but somewhat dense. Presenting results as plots of performance versus compression ratio could improve readability and highlight key trends more clearly.
- The proposed method currently underperforms relative to certain baselines, such as R-KV.
- While the evaluation across multiple model sizes (Figure 3) is useful, presenting six separate figures for this analysis may be excessive. A broader evaluation across more datasets and compression ratios might offer greater impact.

Clarification of the main claim
- The paper’s claim that "the current query contains sufficient information to estimate the importance of all historical tokens" (L157) would benefit from further clarification. This assumption appears to hold primarily in a single-query setting, as discussed in “KVzip: Query-Agnostic KV Cache Compression with Context Reconstruction” (2025, NeurIPS). The query-dependent method estimates token importance only with respect to a specific query.

Need for illustrative figures for the main method (Sec. 3.2)
- The paper could be strengthened by including a clear diagram that visually outlines the proposed method in Section 3.2.

Need for clarification of Figure 2 (Fused Kernel)
- The description of the eviction process in Figure 2 is somewhat difficult to follow. Adding a more detailed diagram regarding the "Compute & Evict Kernel" in the figure would make the main contribution easier to understand. Also, while the middle and right subfigures appear similar to the FlashAttention diagram, the distinctions are not explicitly emphasized.

**Questions:**

Please refer to the weakness section above.

---

### Official Review · Reviewer_NtV1 · 2025-10-30

**Soundness:** 3
**Presentation:** 3
**Contribution:** 2
**Rating:** 4
**Confidence:** 4

**Summary:**

LongFlow introduces a new importance metric for KV cache eviction, and a fused attention kernel implementation that efficiently computes this metric. The importance metric uses only the current query and intermediate values of the standard attention calculation, enabling an efficient kernel. LongFlow achieves 3.67x more throughput than previous work, while benchmark accuracy remains comparable (although not state-of-the-art).

**Strengths:**

1. The contribution of this paper is very practical, providing an efficient kernel implementation which ML practitioners may be eager to adopt. The authors made extensive engineering efforts to optimize the kernel and make it production-ready.
2. LongFlow’s maximum throughput is 3.67x more than previous work (H2O), which is a big efficiency improvement.
3. The paper’s focus on a single query for importance estimation and approximation of the next query with the current query are empirically justified by Figure 5.

**Weaknesses:**

1. LongFlow is not state-of-the-art in benchmark performance, having worse accuracy than R-KV. Although LongFlow has 3.3x more throughput and 25% less memory usage than R-KV, it is not clear when LongFlow would be preferred over R-KV.
2. The experiments conflate the efficiency impact of LongFlow’s new importance metric with the engineering efforts made in the custom Triton kernel (memory pre-allocation, eliminating the outer loop over the query dimension, eliminating running maximum). The efficiency gains of the LongFlow metric should be ablated from the custom kernel by evaluating the LongFlow metric using an existing kernel.
2a. Eliminating the outer loop over the query dimension in FlashAttention limits the extensibility of the kernel. It is especially important to ablate this and understand if this is a major contributor of efficiency.
3. LongFlow’s proposed value rests on its efficiency (since its benchmark accuracy is not better), but its efficiency was only evaluated on a single model with fixed input length, output length, and cache budget.
4. The theoretical derivation of the LongFlow importance metric contains an unjustified simplification: The original global objective of minimizing impact on logits of all future generation steps is simplified to the immediate next generation step. These are very different objectives. Also, the derivation assumes that we should evict 1 token at each step, which is an assumed constraint that is not necessary for all methods. These issues undercut the paper’s theoretical argument.

**Questions:**

Questions:
1. Table 1 shows that the Vanilla performance is sometimes worse than the compressed methods, especially in AIME24 with DeepSeek-R1. This is unexpected since Vanilla “serves as the accuracy upper bound”. How is this possible?
2. In the efficiency experiments, compression is conducted every step for LongFlow but every 128 steps for other methods. Why is it conducted every 128 steps for other methods?

Suggestions:
1. Additional experiments on throughput and memory should be conducted with varying models, model sizes, input/output lengths, and cache budgets to show that LongFlow’s efficiency gains generalize across these factors.
2. The related works on KV Cache Compression should include KVZip[1], a recent KV cache eviction method that outperforms previous methods.

Minor comments:
1. Table 1 should state the unit used for measuring performance.
2. The Vanilla performance in Table 1 should be identical between different budgets, but differ by 0.01 in DeepSeek-R1 GSM8K, and Avg for both models.
3. Efficiency data (e.g. Figure 4) should be provided in a table (perhaps in appendix) for easier numerical comparison.

[1] Kim et al. “KVzip: Query-Agnostic KV Cache Compression with Context Reconstruction.” NeurIPS 2025.

---

### Official Review · Reviewer_wUq4 · 2025-10-31

**Soundness:** 3
**Presentation:** 2
**Contribution:** 2
**Rating:** 4
**Confidence:** 5

**Summary:**

This paper presents LongFlow, a lightweight KV cache compression algorithm specifically designed for reasoning models with long decoding outputs.

It introduces an efficient importance estimation metric that is computed directly from intermediate attention results using only the current query, incurring negligible computational overhead.
The authors also implement a custom Triton kernel that fuses FlashAttention, importance estimation, and token eviction into a single optimized operator.

Experimental results show that LongFlow achieves up to an 11.8× throughput improvement and an 80% reduction in KV cache size, with only minor loss in model accuracy.

**Strengths:**

LongFlow introduces a novel and lightweight importance metric for token eviction, which is directly derived from intermediate attention weights and value vector norms without additional storage overhead.
It also implements an Triton kernel that fuses FlashAttention, importance estimation, and token eviction.

**Weaknesses:**

While LongFlow achieves notable efficiency improvements, its accuracy degradation is non-trivial, particularly on AIME24/25, AMC, and GPQA benchmarks with the Qwen3-8B model.
This degradation may stem from the inherent limitations of KV eviction methods and the **aggressive approximations** LongFlow applies to query vectors and attention denominators.
Due to the existence of **outlier key channels** (where $\max_j |k^j|$ is large) \[1\], the theoretical error bound can grow substantially, undermining eviction accuracy.
Moreover, LongFlow **omits the running maximum of $qk$ scores** in its FlashAttention-based kernel, which weakens **softmax numerical stability** and can further degrade attention precision.

In addition, the **efficiency experiments are limited to Qwen3-1.7B**, and the scalability of the efficiency improvements to larger models remains unverified.

\[1\] Liu et al., KIVI: A Tuning-Free Asymmetric 2bit Quantization for KV Cache

**Questions:**

1. Could the authors **provide an ablation study** on the accuracy loss caused by each approximation in LongFlow, including the **query approximation**, **attention denominator simplification**, and **omission of the running maximum in $qk$ scores**?

2. If these approximations are shown to introduce substantial errors, could the method be **enhanced by incorporating correction mechanisms**?

3. Is **omitting the running maximum of $qk$ scores** truly necessary for efficiency? Have the authors evaluated the trade-off between numerical stability and performance?

4. What are the **evaluation metrics and decoding settings** used in the accuracy experiments (e.g., _pass@k_, temperature, top-p, sampling strategy)?

---

### Official Review · Reviewer_KU4L · 2025-11-01

**Soundness:** 3
**Presentation:** 3
**Contribution:** 2
**Rating:** 4
**Confidence:** 4

**Summary:**

The paper proposes a hardware-efficient KV cache eviction method for reasoning models, along with an optimized fused kernel implementation. To enable this fused kernel, the authors define an importance metric that can be computed directly from intermediate results within the attention computation process. Experimental results show a larger accuracy drop compared to existing baselines, but achieve up to 4× higher throughput.

**Strengths:**

The paper presents a KV cache eviction method specifically designed for reasoning model inference, complemented by a hardware-aware fused kernel implementation that integrates importance computation and KV eviction directly into the attention computation process.

**Weaknesses:**

- The paper is lacking the real end-to-end inference time acceleration within modern inference frameworks such as vLLM or SGLang which is one of the main motivation for proposing the method. A system-level optimization is meaningful only if it demonstrates measurable speedups under realistic deployment environments.
- The current results do not show that the proposed method Pareto-dominates existing baselines. Instead, it appears to present a trade-off solution; achieving higher throughput and lower memory usage at the cost of reduced accuracy compared to R-KV. It would strengthen the paper to include a Pareto analysis by sweeping the compression or reasoning budget, demonstrating whether the proposed method can simultaneously achieve (1) higher throughput, (2) lower memory usage, and (3) higher accuracy than existing methods.
- A simple baseline that needs comparison is just assigning reasoning budgets of 2400 and 3200 tokens for the vanilla model. If the reasoning is not complete, one could insert stop-thinking message such as `Considering the limited time by the user, I have to give the solution based on the thinking directly now.\n</think>.\n\n` and generate the final answer and see the resulting score [1]. It would be interesting to see whether the proposed KV cache eviction methods can even outperform such a simple truncation-based baseline.

[1] Qwen3 Technical Report, 2025, Yang et al.

**Questions:**

- How many tokens are evicted during each decoding phase? Is eviction triggered only after reaching the memory budget? Before hitting the budget, are new tokens simply appended to the cache?
- In Table 1, does the “Vanilla” score correspond to a reasoning budget of 16,000 tokens?
- When performing importance evaluation and eviction for the baselines every 128 steps, how is eviction handled for H2O and VATP? Are 128 tokens evicted at once after exceeding the budget, and are tokens simply appended before the budget limit is reached?

**Details Of Ethics Concerns:**

I do not have ethics concern regarding this paper.

---

### Note · Authors · 2025-12-11

I have read and agree with the venue's withdrawal policy on behalf of myself and my co-authors.